A method for histopathological study of the multifocal nature of spinal cord lesions in murine experimental autoimmune encephalomyelitis

Gibson-Corley Katherine N. katherine-gibson-corley@uiowa.edu
Boyden Alexander W.
Leidinger Mariah R.
Lambertz Allyn M.
Ofori-Amanfo Georgina
Naumann Paul W.
Goeken J. Adam
Karandikar Nitin J.
Department of Pathology, University of Iowa , Iowa City, Iowa , United States
Esteban María Ángeles
Electronic publication date: 2016 Jan 26
Publication date: 2016
Volume: 4
Electronic Location ID: e1600
Received 2015 Nov 25; Accepted 2015 Dec 23
Copyright: © 2016 Gibson-Corley et al.
Copyright year: 2016
Copyright holder: Gibson-Corley et al.
License: This is an open access article distributed under the terms of the Creative Commons Attribution License, which permits unrestricted use, distribution, reproduction and adaptation in any medium and for any purpose provided that it is properly attributed. For attribution, the original author(s), title, publication source (PeerJ) and either DOI or URL of the article must be cited.
License URL: https://creativecommons.org/licenses/by/4.0/

Keywords: Mice, Histopathology, Spinal cord, EAE

Funding: This work was supported, in part, by grant awards to NJK from the NIH and National MS Society. The funders had no role in study design, data collection and analysis, decision to publish, or preparation of the manuscript.

==============================
Experimental autoimmune encephalomyelitis (EAE) is a well-established mouse model for multiple sclerosis and is characterized by infiltration of mononuclear cells and demyelination within the central nervous system along with the clinical symptoms of paralysis. EAE is a multifocal and random disease, which sometimes makes histopathologic analysis of lesions difficult as it may not be possible to predict where lesions will occur, especially when evaluating cross sections of spinal cord. Consequently, lesions may be easily missed due to limited sampling in traditional approaches. To evaluate the entire length of the spinal cord while maintaining anatomic integrity, we have developed a method to section the cord within the decalcified spinal column, which allows for the study of the multifocal nature of this disease and also minimizes handling artifact. HE and Luxol fast blue staining of these spinal cord sections revealed a paucity of lesions in some areas, while others showed marked inflammation and demyelination. The percentage of spinal cord affected by EAE was evaluated at four separate areas of longitudinally sectioned cord and it varied greatly within each animal. Immunohistochemical staining of in situ spinal cords which had undergone decalcification was successful for key immuno-markers used in EAE research including CD3 for T cells, B220 for B cells and F4/80 for murine macrophages. This method will allow investigators to look at the entire spinal cord on a single slide and evaluate the spinal cord with and without classic EAE lesions.

Introduction

Multiple sclerosis (MS) is a debilitating autoimmune disease characterized by cellular inflammation into–and the progressive demyelination of–the central nervous system (CNS), subsequently leading to a multitude of clinical symptoms (Sospedra & Martin, 2005; Steinbach & Merkler, 2014; Trapp & Nave, 2008). Experimental autoimmune encephalomyelitis (EAE) is a well-studied mouse model of MS-like disease due not only to its convenience for immune system manipulation, but importantly because it recapitulates various hallmarks of human disease such as CNS inflammation and paralysis (Constantinescu et al., 2011; Duffy, Lees & Moalem-Taylor, 2014; McCarthy, Richards & Miller, 2012).

EAE can be induced by an assortment of immunizations, but most often includes the subcutaneous injection of peptides derived from neuroantigens (such as myelin basic protein, myelin proteolipid protein, or myelin oligodendrocyte glycoprotein (MOG)) as part of an emulsion with Complete Freund’s Adjuvant (CFA). In this study, we induced EAE disease in C57BL/6 mice with the commonly utilized MOG35–55 peptide. Upon disease onset of this classic model, the manifestation of clinical symptoms ensues, which is characterized by a grading scale of ascending paralysis (Ortega et al., 2013; York et al., 2010) (Table 1). Inflammatory cell infiltration and demyelination of the CNS in MS/EAE drive clinical disease symptoms and therefore histological assessment of the spinal cord can be crucial in evaluating EAE disease, as well as any past and future therapeutics in this model. Indeed, researchers’ ability to efficiently and accurately evaluate MS-like disease within the EAE model has been and will continue to be productive in pushing the MS and CNS inflammatory disease fields forward.

Table 1 Clinical scoring rubric for EAE.

Score	Clinical findings	
1	loss of tail tonicity	
2	mild hind limb weakness	
3	partial hind limb paralysis	
4	complete hind limb paralysis	
5	complete hind limb paralysis with forelimb weakness or moribund	

Key histopathologic findings associated with EAE include inflammatory cell infiltration and axonal loss (Steinbach & Merkler, 2014) which can be identified using routine techniques (Klopfleisch, 2013). These include hematoxylin and eosin (HE) and Luxol fast blue (LFB) stains as well as immunohistochemical staining for important inflammatory cell markers such as T cells, B cells and macrophages (Steinbach & Merkler, 2014). This type of histopathologic analysis is commonly performed on EAE-diseased murine spinal cord, which is classically sectioned in a cross-wise (coronal) fashion. While this allows researchers to visualize the entire cord including both white and gray matter, it is at a single level and only approximately 5 μm thick. This type of sectioning can be and often is enlightening, however the vast number of spinal cord cross sections makes examination cumbersome. Furthermore, EAE is a multifocal and random disease (Day, 2005) and thus it is impossible to predict where lesions will occur throughout the spinal cord, even within areas of expected lesion localization. To histologically evaluate the entire murine spinal cord, we have developed a method to section the cord longitudinally within the decalcified spinal column allowing us to identify and study the multifocal nature of EAE disease.

Materials and Methods

Mice

C57BL/6 mice (females, 6–8 weeks old) were purchased from The Jackson Laboratory (Bar Harbor, ME, USA). Mice were allowed chow and water ad libitum, maintained on a 12-hour light/dark cycle, and housed in specific pathogen-free barrier facilities at the University of Iowa. All animal and tissue work was approved by the University of Iowa Institutional Animal Use and Care Committee.

Immunizations and EAE evaluation

On day 0, mice were immunized subcutaneously with 50 μg of a myelin oligodendrocyte glycoprotein peptide (MOG35–55) emulsified 1:1 volume in CFA supplemented with 4 mg/mL Mycobacterium tuberculosis (H37Ra; Difco Laboratories, Detroit, MI, USA). Mice were additionally injected intraperitoneally on days 0 and 2 with 250 ng of pertussis toxin (List Biological Laboratories, Campbell, CA, USA). Clinical EAE disease scores were monitored using the grading scale as follows: 1) loss of tail tonicity; 2) mild hind limb weakness; 3) partial hind limb paralysis; 4) complete hind limb paralysis; 5) complete hind limb paralysis with forelimb weakness or moribund/death (Table 1). Specific mice were chosen at various scores for histological evaluation.

Tissue preparation and histology

Mice were humanely euthanized by carbon dioxide asphyxiation in accordance with NIH & ACURF guidelines. Rapid fixation was achieved by whole body perfusion with the use of a simple gravity flow device utilizing 10% neutral buffered formalin (NBF) (Leica Biosystems, Wetzlar, Germany). A 60 cc syringe barrel with attached stopcock & 3 mm diameter tubing was mounted on a ring stand with the syringe 80 cm above the working surface. The syringe and tubing were flushed with 37 °C PBS prior to and after each use. Following euthanasia, the thoracic cavity was immediately opened exposing the heart, the right atrium severed, and a 22 g needle attached to the gravity flow tubing inserted into the left ventricle. Flow was initiated and the animal perfused with an initial 5 ml of PBS at 37 °C to clear the vasculature of blood followed by immediate perfusion with 40 ml of 10% NBF at 37 °C.

Once perfused, the entire spinal column, including the vertebrae and enclosed spinal cord, were removed, epaxial muscles dissected off, and placed in 200 ml of 10% NBF for 3 days at room temperature on an orbital shaker set at 100 RPM for immersion fixation. After 3 days in NBF, the spines were briefly washed with tap water and placed in 200 ml of 14% EDTA (Sigma ED-EDTA, pH 7.3) for decalcification with continuous shaking. The spines were in 14% ED-EDTA for 4 days before removal, washed thoroughly with tap water for 3 hours, and sections grossed into cassettes with the use of a microtome blade. The entire spinal column was sectioned in half into longitudinal sections thus exposing the centrally located spinal cord and marking dye was used on the samples to maintain appropriate orientation. Tissues were placed back into 10% NBF, routinely processed, embedded in paraffin, and consecutive sections at 5 μm thickness were cut for subsequent staining (Fig. 1).

Figure 1 Process of tissue collection and histologic preparation of longitudinally sectioned in situ spinal cord sections.

Routine HE and Luxol fast blue (LFB) staining was performed on all sections. Digital images were collected with a DP73 camera and CellSens software (Olympus, Tokyo, Japan). HE-stained, longitudinally sectioned spinal cord sections were evaluated for lesions of EAE, including demyelination and inflammatory cell infiltration. Four separate areas along each spinal cord were identified and in each area the percentage of spinal cord with lesions was estimated visually (using a scale of 0, 10, 20….100% affected) at 20× magnification by a board-certified veterinary pathologist.

Immunohistochemistry

To validate that immunohistochemistry would be successful using this method of tissue preparation, staining was performed for key inflammatory cells common in EAE (CD3 for T cells, B220 for B cells and F4/80 for macrophages) (Table 2). All immunohistochemical staining was performed manually using peroxidase methods and Dako Envision systems (Glostrup, Denmark).

Table 2 Primary antibodies and their commercially available sources, catalog numbers, dilutions and specific antigen retrieval conditions utilized in the study.

Marker	Antibody	Dilution	Source	Conditions	
CD3	Cat# RM-9107-5	1:200	Neomarkers	HIER, citrate buffer (pH 6.0)	
B220	Cat# MCA1258G	1:6000	Serotec	HIER, citrate buffer (pH 6.0)	
F4/80	Cat# MCAP497	1:6400	Serotec	HIER, citrate buffer (pH 6.0)	

Results

HE and LFB staining

HE and LFB staining of longitudinally sectioned spinal cord sections (Fig. 2) revealed the multifocal to coalescing nature of the lesions associated with EAE. In some areas, the spinal cord can appear relatively normal while in other areas there is significant pathology (Figs. 2A and 2D). When a section of spinal cord from a similarly affected mouse is cut cross-wise, it can be taken from an area with a paucity of lesions (Figs. 2B and 2E), this would lend an investigator to think the cord was relatively unaffected. In contrast, when a cross section is taken within an area of pathology, which includes infiltration of inflammatory cells and demyelination (Figs. 2C and 2F), the severity of disease could be overestimated.

Figure 2 Photomicrographs of decalcified spinal columns from EAE mice.

Longitudinally sectioned spinal cord stained with HE (A) and LFB (D) showing an area which is relatively unaffected (left dotted line) versus one with significant demyelination and inflammatory cell infiltration (right dotted line). The relatively unaffected area corresponds to the HE (B) and Luxol Fast Blue (E) cross sections and the area with lesions corresponds to the HE (C) and Luxol Fast Blue (F) cross sections. Solid arrows indicate areas of inflammation and dotted arrows indicate areas of demyelination. Bars = 200 μm.

Prior to euthanasia, each animal was given a clinical score based on their symptoms of EAE (Table 1). The clinical scores ranged from 2 (mild hind limb weakness) to 4 (complete hind limb paralysis) (Fig. 3A). Following necropsy, the percentage of spinal cord affected by EAE lesions was identified at four separate and equidistant areas of the longitudinally sectioned, HE-stained spinal cord sections to determine the variation in lesion severity. Interestingly, the percentage of spinal cord affected by EAE lesions varied greatly—even within a single sample and that, as expected, the most severe lesions appear to be in more caudal spinal cord (Fig. 3B) (Nathoo, Yong & Dunn, 2014). This illustrates the utility of longitudinally sectioning EAE spinal cords to increase the odds of identifying lesions.

Figure 3 Clinical EAE score and quantification of the percentage of spinal cord affected by EAE using the method of longitudinal sectioning.

(A) Clinical EAE score (Table 1) of 12 different mice with EAE ranging in age from 7–11 weeks of age. (B) Four separate areas (divided equally cranial to caudal) of longitudinally sectioned spinal cord from the same 12 mice, which were evaluated to determine what percentage of the cord at that area was affected by EAE lesions.

Immunohistochemical staining

Immunohistochemical staining was performed on these longitudinally sectioned, decalcified specimens (Fig. 4). Specific immunomarkers (Table 2) commonly used in EAE research were selected, including CD3 for T cells (Fig. 4B), F4/80 for macrophages (Fig. 4C) and B220 for B cells (Fig. 4D). All successfully stained the decalcified tissues and were able to identify these key immune cells within the EAE spinal cord lesions.

Figure 4 Representative images of key immunohistochemical markers run on a longitudinally sectioned, decalcified spinal column from an EAE mouse with a clinical EAE score of 4.

(A) HE, (B) B220 immunohistochemistry for (B) cells, (C) F4/80 immunohistochemistry for macrophages (arrow indicates area highlighted in inset, inset bar = 20 μm), (D) CD3 immunohistochemistry for T cells. Bars = 200 μm.

Discussion

Due to the multifocal nature of EAE within the spinal cord, we developed a novel way of looking at the entire length of the spinal cord in situ, within the vertebrae, allowing us to visualize the multifocal nature of lesions (Fig. 1). Using both HE and LFB staining, we show that in some areas of the spinal cord there can be significant pathology but in other areas the cord can appear relatively normal (Fig. 2). The histopathologic score of the spinal cords somewhat followed the clinical score of these mice antemortem (Fig. 3). Interestingly, some animals (mouse 1 and 11) had markedly variable histopathology but had fairly high clinical scores, indicating that clinically there was significant damage to the cord that might be difficult to identify histologically if one did not section the area(s) of cord affected. It is also important to note that this technique, which uses a decalcification step, also allows for routine immunohistochemical staining for key inflammatory cell infiltrates. Decalcification can affect some immunohistochemical assays but in this case we were able to assess for the key immune cells in EAE; T cells, B cells and macrophages (Fig. 4) (Bussolati & Leonardo, 2008).

This technique will allow EAE researchers to quickly assess the spinal cord using routine histopathology and without having to make numerous slides of spinal cord cross sections. It is also important to note that the spinal cord itself does not have to be flushed from the spinal column, which not only saves time, but also preserves the integrity of the cord and minimizes handling artifacts within the tissue. Of course, there are limitations. With longitudinal sectioning, one can only visualize a single plane of the cord unlike cross sectioning where all of the white and gray matter can be assessed. If the histotechnologist does not cut deep enough into the tissue, only the white matter will be visible as the deeper gray matter can be missed. It is also difficult to visualize the central canal using longitudinal sectioning. Another challenge is working with the decalcified vertebrae surrounding the softer spinal cord. The spinal column isn’t completely flat so the undulating nature of the vertebrae can pose challenges to the histotechnologist during cutting to make sure the spinal cord isn’t folded or shredded.

In classic EAE the disease presents with hind limb paralysis and the lesions are predominantly seen in the lumbosacral region of the spinal cord, as demonstrated in Fig. 2. However, it is noteworthy that there are lesions in thoracic and higher segments, which are generally missed in routine evaluation. Moreover, histologic evaluation of EAE using this novel approach will add tremendous value in understanding models of atypical EAE (such as those in certain knockout mice) or after therapeutic intervention in classic EAE, where certain lesions may resolve and/or re-myelinate, whereas others may not. This is a current gap in the field that could benefit from the methods presented here.

Conclusions

In summary, longitudinal sectioning of the decalcified spinal column in mice is a useful technique to study the multifocal and oftentimes random nature of spinal cord lesions encountered in EAE. This technique will be useful to the researcher who would like to assess the entire length of the spinal cord without having to remove the spinal cord from the vertebrae and without having to make numerous spinal cord cross sections.

Supplemental Information

Supplemental Information 1 Raw data for Figure 3.

Raw Data for Figure 3.

Click here for additional data file.

Additional Information and Declarations

Competing Interests

Author Contributions

Animal Ethics

Data Deposition

The authors declare that they have no competing interests.

Katherine N. Gibson-Corley conceived and designed the experiments, performed the experiments, analyzed the data, contributed reagents/materials/analysis tools, wrote the paper, prepared figures and/or tables, reviewed drafts of the paper.

Alexander W. Boyden performed the experiments, analyzed the data, contributed reagents/materials/analysis tools, wrote the paper, prepared figures and/or tables, reviewed drafts of the paper.

Mariah R. Leidinger performed the experiments, contributed reagents/materials/analysis tools, reviewed drafts of the paper.

Allyn M. Lambertz performed the experiments, contributed reagents/materials/analysis tools, reviewed drafts of the paper.

Georgina Ofori-Amanfo performed the experiments, contributed reagents/materials/analysis tools, reviewed drafts of the paper.

Paul W. Naumann performed the experiments, contributed reagents/materials/analysis tools, reviewed drafts of the paper.

J. Adam Goeken performed the experiments, contributed reagents/materials/analysis tools, reviewed drafts of the paper.

Nitin J. Karandikar conceived and designed the experiments, analyzed the data, contributed reagents/materials/analysis tools, wrote the paper, prepared figures and/or tables, reviewed drafts of the paper.

The following information was supplied relating to ethical approvals (i.e., approving body and any reference numbers):

All animal and tissue work was approved by the University of Iowa Institutional Animal Use and Care Committee (#1301014).

The following information was supplied regarding data availability:

The research in this article did not generate any raw data.

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
