# Peer review of "A method for histopathological study of the multifocal nature of spinal cord lesions in murine experimental autoimmune encephalomyelitis"

_PeerJ, doi:10.7717/peerj.1600_

## Round 0.1 · original submission · Minor Revisions

Please, amend your manuscript according to the suggestions.

Reviewer 1 ·

Basic reporting

Excellent

Experimental design

Excellent

Validity of the findings

Excellent

Additional comments

As excellent as the experimental design and report of results are, there is no novel observation in the study. The technique of observing the whole spinal cord seems good but the question is why would anyone do that when these rodents present mainly with hindlimb paralysis and any infiltration in the thoracic or higher segments of the spinal cord would be accompanied by a more severe disease. And so the technique, as excellently demonstrated as the authors have presented it, does not add anything to the state of knowledge of the basis or study of disease process. Also the demonstration demyelination is not impressive but that is not surprising with the model of disease used. I am unable to reject this article for the reasons given below because the study is not fundamentally flawed.

Reviewer 2 ·

Basic reporting

The authors describe a method of sectioning the spinal cord within the decalcified spinal column, allowing for the study of the entire spinal cord to provide for a more comprehensive assessment of spinal cord pathology. The authors described the limitations and they are applauded for doing so. The work should be of interest to many in the EAE field. The following are points of clarification:

1. While it is clear from the description and from figure 1 that the entire length of the spinal cord can be viewed, what is not clear is whether the authors then made consecutive longitudinal sections through the length of the cord (which I presume to be the case) and the thickness of each section.
2. What is also not clear is how coronal sections (Figure 2b) were obtained in their entirety when the cord has been longitudinally sectioned through.
3. In Figure 3, the scale showing cranial to caudal should be inverted as more of the cranial results are in the lower part of the panel, and vice versa.

Experimental design

Well done

Validity of the findings

Well done

Additional comments

See the basic reporting section above

---

## Round 0.2 · accepted · Accept

The manuscript has been improved as indicated.